# Advanced Polymers for Three-Dimensional (3D) Organ Bioprinting

**DOI:** 10.3390/mi10120814

**Published:** 2019-11-25

**Authors:** Xiaohong Wang

**Affiliations:** 1Center of 3D Printing & Organ Manufacturing, School of Fundamental Sciences, China Medical University (CMU), No. 77 Puhe Road, Shenyang North New Area, Shenyang 110122, China; wangxiaohong709@163.com or wangxiaohong@tsinghua.edu.cn; Tel./Fax: +86-24-31900983; 2Center of Organ Manufacturing, Department of Mechanical Engineering, Tsinghua University, Beijing 100084, China

**Keywords:** three-dimensional (3D) printing, organ manufacturing, biomaterials, polymers, stem cells

## Abstract

Three-dimensional (3D) organ bioprinting is an attractive scientific area with huge commercial profit, which could solve all the serious bottleneck problems for allograft transplantation, high-throughput drug screening, and pathological analysis. Integrating multiple heterogeneous adult cell types and/or stem cells along with other biomaterials (e.g., polymers, bioactive agents, or biomolecules) to make 3D constructs functional is one of the core issues for 3D bioprinting of bioartificial organs. Both natural and synthetic polymers play essential and ubiquitous roles for hierarchical vascular and neural network formation in 3D printed constructs based on their specific physical, chemical, biological, and physiological properties. In this article, several advanced polymers with excellent biocompatibility, biodegradability, 3D printability, and structural stability are reviewed. The challenges and perspectives of polymers for rapid manufacturing of complex organs, such as the liver, heart, kidney, lung, breast, and brain, are outlined.

## 1. Introduction

In nature, all the complicated phenomena of life, including human organs, are the result of biochemical and biophysical changes of molecules (or materials at a molecular level). For example, small organic molecules or compounds combine to form larger polymers (macromolecules or biomacromolecules). Macromolecules arrange in specific ways to form cells with organelles inside the cell membrane (Figure 1). While homogeneous cells organize into tissues, heterogeneous cells aggregate into organs with particular physiological functions. It has taken, in some cases, thousands of years to evolve from tiny organic molecules to microcells, mesotissues, and macro-organs [1,2,3].

At present, organ failure is the main cause of mortality all over the world. Despite the rapid development in pharmacological, interventional, and surgical therapies during the last several decades, the only cure for organ failure is allograft organ transplantation, which is seriously limited by issues, such as donor organ shortage, life-long immune rejection, and ethical conflict [4,5,6].

Three-dimensional (3D) organ bioprinting is the utilization of 3D printing technologies to assemble multiple cell types or stem cells/growth factors along with other biomaterials in a layer-by-layer fashion to produce bioartificial organs that maximally imitate their natural counterparts [7,8,9]. Traditionally, 3D printing is named rapid prototyping (RP), solid freeform fabrication (SFF), or additive manufacturing (AM) based on the dispersion–accumulation (i.e., discrete–accumulation) principle of computer-aided manufacturing (CAM) techniques. Before 3D printing, an object can be divided into numerous two-dimensional (2D) layers with a defined thickness. These 2D layers can be piled up by selectively adding the desired materials in a highly reproductive layer-by-layer manner under the instruction of computer-aided design (CAD) models [10,11,12,13]. Patient-specific organ image data, such as computerized tomography (CT) and magnetic resonance imaging (MRI), can be easily transferred into CAD models for customized organ manufacturing with predefined geometrical shapes, biomaterial constituents, and physiological functions [14,15,16,17].

The 3D organ bioprinting procedure involves changes to the properties of a series of materials at molecular, cell, tissue, and organ levels. It is an emerging new interdisciplinary field that needs cooperation of many fields of science and technology, such as biomaterials, biology, physics, chemistry, computers, mechanics, bioinformatics, and medicine (Figure 2).

During the last 16 years, a large variety of 3D bioprinting technologies have been exploited, which has led to the emergence of fully automatic manufacturing of bioartificial organs for wide biomedical applications, such as high-throughput drug screening, controlled cell transplantation, customized organ repair/regeneration/replacement/restoration, pathological mechanism analysis, metabolism model establishment, and living tissue/organ cryopreservation [10,11,12,13,14,15,16,17,18,19,20]. Based on the working principles, these technologies can be classified into four major groups: inkjet-based, extrusion-based, laser-based, and their combinations (Figure 3). Each of the former three groups has advantages and disadvantages in bioartificial organ manufacturing.

Several series of unique automatic and semiautomatic bioartificial organ manufacturing technologies have been created in my own group with the proper integration of modern high technologies, including computer, biology (e.g., cells and stem cells), biomaterials (e.g., polymers), chemistry, mechanics, and medicine. With these unique high technologies, we have solved all the bottleneck problems that have perplexed tissue engineers and other scientists for more than 6–7 decades, such as large-scale tissue/organ manufacturing [21,22,23,24], hierarchical vascular/nerve network construction with a fully endothelialized inner surface and antisuture/antistress capabilities [25,26,27,28,29], step-by-step adipose-derived stem cell (ASC) differentiation in a 3D construct [30,31,32], long-term preservation of bioartificial tissues/organs [33,34,35], in vitro metabolism model establishment [36,37], high-throughput drug screening [38,39,40], in vivo biocompatibilities of implanted biomaterials [41,42,43]. Several polymers have played essential and ubiquitous roles for bioartificial organ manufacturing with the incorporation of multiple cell types, stem cells/growth factors, and hierarchical vascular and neural networks with antisuture and antistress functions.

## 2. Role of Polymers in 3D Organ Bioprinting

3D organ bioprinting is not a simple and easy engineering approach. Like building a nuclear plant, it requires intermingling of intricate architectural design, appropriate biomaterial selection, special building process, multicellular incorporation, supportive structure utilization, controllable stem cell induction, simultaneous or sequential tissue formation and maturation, multiple tissue coordination, and the means of coordinating these procedures to form large-scale living organs [25,26,27,28,29,30,31,32,33].

Polymers are large molecules made up of many small and identical repeating units bonded by covalent bonds [44]. The smallest repeating unit in a polymer is called a monomer, while the number of repeat units in a polymer chain is termed the degree of polymerization (DP) or chain length. Polymers can be divided into natural and synthetic groups according to their origin. Most natural polymers are water-soluble and endowed with some common biological and physiological properties, such as being pliable as soft tissues and organs, friendly for cell encapsulation and transplantation, and easy to handle and reshape. The properties of synthetic polymers depend on many factors, such as processing conditions, molecular weights, monomer distributions, chain structures (e.g., size, geometry, inter/intrabonding, and branching), and the presence of additives. In particular, linear polymers have long chains (or backbones) that contain small chemical groups on the repeating units.

Hydrogels are 3D hydrophilic networks of polymers that can absorb and retain large amounts of water and gel under certain physical (e.g., thermosensitive), chemical (e.g., covalent bonding), or biochemical (e.g., enzymatic) conditions. The physical and chemical properties of hydrogels can be designed for specific biomedical applications by selecting proper polymer components, inorganic solvents, and gelation protocols [45,46,47]. Compared with other states of polymers, hydrogels can provide a benign and stable environment for living cells to grow, migrate, aggregate, proliferate, and differentiate inside. The integration of hydrogels with 3D bioprinting technologies has offered numerous attractive features for complex organ manufacturing [48,49,50].

During the 3D organ bioprinting process, different polymers have different roles and functions. Most natural polymers, which are employed as the main components of bioinks, have the following roles: (1) provide cells and bioactive agents with support such as accommodation; (2) build vascular, neural, and lymphatic networks as semipermeatable substrates for nutrient, gas (e.g., oxygen), metabolite, and biosignal exchange; (3) guide homogeneous and heterogeneous histogenesis and organogenesis in a predefined way; and (4) promote tissue and organ maturation under specific biochemical and biophysical conditions. Meanwhile, most synthetic polymers are applied for the following functions: (1) improve multicellular handling or allotting in space to mimic their natural counterparts; (2) enhance the mechanical properties of vascular and neural networks with antisuture and antistress capabilities; (3) commit (or complete) extra functions, such as sacrificing supports and protecting covers.

Globally, the literature has reported on a number of polymers for use in 3D bioprinting. These are summarized in Figure 4 [51,52,53,54,55,56,57,58,59,60,61,62,63,64,65,66,67,68,69,70,71,72,73,74]. These polymers need to meet several basic requirements for 3D printing of bioartificial organs and subsequently clinical applications (Figure 5): (1) biocompatible (i.e., nontoxic or no obvious toxicity, no or low immunological reaction); (2) bioprintable using 3D bioprinters; (3) biostable or crosslinkable with strong enough mechanical properties; (4) biodegradable (in particular, the biodegradation rate should match the new tissue/organ generation speed); (5) suturable with host vascular and nerve networks; (6) permeable for nutrients and gases; (7) biostorable before being printed; and (8) sterilizable.

In the following sections, seven normal polymers that have been frequently employed in 3D organ bioprinting with excellent biocompatibility, biodegradability, biostability, and bioprintability are individually analyzed according to their natural or synthetic properties.

## 3. Natural Polymers for 3D Organ Bioprinting

Natural polymers are widely existent in animal, plant, and microbe tissues as the main components of extracellular matrices (ECMs) or decellularized extracellular matrices (dECMs). These polymers include (1) proteins, such as collagen I–V, elastin, keratin, actin, tubulin, and myosin; (2) polysaccharides, such as chitin, alginate, and starch; (3) glycoproteins, such as mucin, lectin, miraculin, transferrin, and nectin; (4) proteoglycans, such as decorin, syndecan, versican, betaglycan, lumican, and fibromodulin. Compared with synthetic polymers, most natural polymers dissolve in inorganic solvents, such as water, phosphate buffer saline (PBS), and Dulbecco’s modified Eagle medium (DMEM), which are cell-friendly. Few of the natural polymer solutions or hydrogels can be used directly as cell-loading matrices for 3D organ bioprinting.

Due to the special physical, chemical, and biological properties, most natural polymer solutions or hydrogels cannot be printed alone with a sol–gel transformation taking place during the 3D printing processes [25,26,27,28,29,30,31,32,33]. These polymers are often used as additives for several theromsensitive or chemical crosslinkable polymer (e.g., gelatin, agar/agarose, and alginate) solutions or hydrogels for 3D bioprinting. The 3D bioprintability of natural polymers are mainly determined by the molecular weight, viscosity, hydrophilicity, and crosslinkability of the polymer solutions or hydrogels. The 3D bioprinting accuracy of the cell-laden natural polymer solutions or hydrogels depends largely on the polymer concentration, viscoelasticity, gelation speed, and shear thinning behavior.

The main advantages of natural polymers for 3D organ bioprinting is that they can entrap viable cells and bioactive agents before printing, protect cells and bioactive agents during 3D printing, and form semipermeable substrates after 3D printing. Before 3D printing, cells and bioactive agents are normally embedded in natural polymer solutions or hydrogels. During 3D printing, the natural polymer chains can safeguard cells from printing stress and provide cells with predesigned 3D milieus similar to those in a native organ. After 3D printing, the polymer chains can be physically/chemically/enzymatically crosslinked to form semipermeable substrates. These semipermeable substrates are permeable to nutrients, gases (e.g., oxygen), and metabolites of cells.

At present, the most frequently used natural polymers for 3D organ bioprinting are collagen, gelatin, alginate, fibrinogen, starch, hyaluronan, chitosan, silk, dextran, agar (or agarose), and matrigel (or dECM). Among these polymers, alginate, gelatin, fibrinogen, and dECM are the most promising candidates for 3D organ bioprinting.

### 3.1. Alginate

Alginate, the salt of alginic acid, is an anionic polysaccharide derived from brown seaweed algae. It consists of β-D-mannuronic acid (M block) and α-L-guluronic acid (G block) monomers in its molecule. Alginate itself can dissolve in water and be crosslinked by divalent cations, such as calcium (Ca^2+^), barium (Ba^2+^), and strontium (Sr^2+^) ions, due to ion exchange reactions. This characteristic is particularly attractive in many biomedical fields, such as nanoparticle preparation, drug delivery, wound healing, tissue engineering, and regenerative medicine [75,76,77,78]. An obvious characteristic of alginate solution is that its physical sol–gel transition point is below 0 °C. Under ambient temperature, it is hard for pure alginate solution to be printed in layers without chemical crosslinking. The in vivo biocompatibility of alginate is not as good as those of animal- or human-derived natural polymers, such as gelatin and fibrinogen [41].

Generally, the physiochemical properties of alginate hydrogels depend on the ratio of M/G blocks. The higher the M/G ratio, the higher the activity of the polymers. The first pertinent alginate 3D bioprinting technology was reported in 2005, in which alginate was used as an additive in gelatin-based cell-laden bioinks (Figure 6) [25,26,27,28,29,30,31,32,33]. The blending of alginate with gelatin molecules can improve the printing resolution and increase the shape fidelity. Only a certain range of the alginate/gelatin concentrations can be printed in layers. Optimal concentration of alginate in gelatin-based bioinks varies from 0.5% (w/v) to 3% (w/v) depending on the polymer resources. After 3D printing, calcium ion crosslinks (i.e., ion bonds) can significantly improve the structural stability to a certain degree. The 3D printed constructs can be crosslinked through various liquid exposures, such as spraying, soaking, and filtering of calcium solutions [79,80,81]. This exposure leads to the exchange of sodium ions in the alginate molecules with Ca^2+^ ions whereby the divalent Ca^2+^ ions form chemical crosslinks or chelates between two carboxyl groups in the same or different polymer chains.

It is very interesting that the chemical crosslinking of alginate molecules using calcium ions is reversible. When the 3D printed constructs are placed in a liquid containing no or less Ca^2+^, the crosslinked Ca^2+^ dissolves gradually within about one week. Further reinforcement is necessary when long-term in vitro culture is required. This means that the 3D constructs need to be further stabilized on and off using calcium ions during long-term in vitro cultures.

Alginate-based 3D bioprinting processes can be completed through different working mechanisms, such as extrusion-based cell-laden fiber deposition on a platform [82], coaxial nozzle-assisted crosslinking deposition, bioplotting in a plotting medium (i.e., crosslinker pool) [83], and precrosslinked alginate hydrogel coextruded with cells [84]. Each of these 3D bioprinting technologies has some merits in 3D printing of bioartificial organs.

In extrusion-based 3D bioprinting, great effort has been made to improve printing resolution and shape fidelity of the cell-laden alginate-containing 3D constructs by optimizing the processing parameters, such as nozzle size, dispensing pressure, and printing speed [85,86,87,88,89]. For example, Markstedt and coworkers blended alginate hydrogel with collagen and nanofibrillated cellulose to effectively enhance extrusion-based 3D printing resolution from 1000 to 400–600 μm using a proper diameter nozzle [90]. Kundu and coworkers printed cell-laden alginate with synthetic polycaprolactone (PCL) using a double-nozzle 3D bioprinter [14].

### 3.2. Gelatin

Gelatin is a partly hydrolyzed collagen derived from different animal tissues, such as bovine tendon and fish. Unlike its ancestor collagen, gelatin is a typical thermal-responsive (or thermosensitive) linear natural polymer with excellent water solubility, biocompatibility, biodegradability, and 3D printability. For example, the immunologic rejection of gelatin is much lower (i.e., zero) than that of collagen. There are no inflammation and other negative reactions when gelatin hydrogels are implanted in vivo [91]. This is an outstanding advantage of gelatin hydrogels for use as bioinks for 3D bioprinting of bioartificial organs.

The liquefaction temperature (i.e., sol–gel transition point) of gelatin solution is approximately 28 °C (25–30 °C). The unique sol–gel transition property and super biocompatibility of gelatin hydrogels have made them the preferential natural bioinks for 3D organ bioprinting. Before 3D bioprinting, cells and bioactive agents are homogeneously encapsulated in the gelatin solution to form cell-laden hydrogels. Other natural polymers, such as alginate, fibrinogen, chitosan, and hyaluronate, can be incorporated into the gelatin solution as additives to form composite bioinks [92,93,94,95,96,97]. Different chemical crosslinking strategies have been employed to improve the stability of 3D printed constructs according to the properties of the additives.

During the sol–gel transition stage, physical crosslinking occurs among the gelatin molecules. This means that thermosensitive gelatin solutions are capable of gelling and being 3D printed at mild or room temperatures, such as 1–28 °C, in which cells are durable. However, physical crosslinking (or gelling) is a reversible thermosensitive gelation process. The bonding strength among gelatin molecules is poor, which results in breakage of the 3D printed constructs when they are put into culture medium at physiological conditions, such as 37 °C. Some groups have utilized this phenomenon to print monolayer cells using extrusion-based 3D printers and washed the liquefied gelatin molecules post printing [98]. Otherwise, secondary chemical crosslinking is necessary to stabilize the 3D constructs [79,80,81].

Since 2005, various gelatin-based composite bioinks, such as gelatin/alginate, gelatin/fibrin, gelatin/chitosan, gelatin/hyaluronate, gelatin/alginate/fibrin, and gelatin/alginate/dextron (glaycerol or dimethyl sulfoxide), have been explored in my laboratory through various extrusion-based 3D bioprinting models (Figure 7) [15,32,79,80,81]. In these models, the viscosity of the gelatin-based bioinks depends largely on the polymer concentration, molecular weight, and cell density. A series of two-step stabilization strategies, containing both thermosensitive physical and ionic chemical crosslinks, have been exploited for 3D printed constructs. The chemical crosslinking methods include glutaraldehyde for gelatin, CaCl_2_ for alginate, sodium tripolyphosphate for chitosan, and thrombin for fibrinogen. Ten years on, these classical bioink formulations and crosslinking strategies have been widely adapted by many other groups all over the world [96,97].

When cells are embedded in gelatin-based solutions with high polymer concentration, their bioactivities are restricted to some degree after the double physical and chemical crosslinking. Meanwhile, lower concentrations of polymers facilitate higher cell viability. Nevertheless, when the concentration of gelatin-based polymers is reduced, the mechanical strength of the 3D constructs drops sharply despite chemical crosslinking. An optimized polymer concentration is necessary for a typical 3D bioprinting technology to ensure favorable cellular activity and structural stability [79,80,81].

Because the components of our gelatin-based composite bioinks are similar to ECMs (i.e., proteoglycans or glycoproteins), 3D printed constructs can be engineered to possess similar water content and elastic modulus as those of soft organs with excellent in vitro cell and in vivo tissue biocompatibilities. This is a fundamental breakthrough in large-scale 3D organ bioprinting. Until now, the resulting 3D printed constructs have been extended rapidly to other biomedical areas, such as high-throughput drug screening, living tissue and organ cryopreservation, metabolism model establishment, and pathological mechanism analysis [79,80,81].

Besides physical and chemical crosslinking strategies, some researchers have aimed to maintain laser-based 3D printed gelatin structures by methacrylation of the gelatin molecules (i.e., gelatin methacrylamide or gelatin methacryloyl, GelMA) using irreversible ultraviolet (UV) photopolymerization techniques [99]. Some researchers have blended GelMA with poly(ethylene glycol) (PEG) and other polymers. It was reported that the extruded strut size could be reduced from 1100–1300 to 350–450 µm by blending GelMA with PEG and to 150–200 µm by coaxial extrusion of GelMA with alginate using a coaxial nozzle [100,101,102,103]. Additional concerns have arisen regarding the toxicity of polymerization initiators and degradation products of synthetic polymethacrylamide.

### 3.3. Fibrin

Fibrin is a blood-derived fibrous protein formed by fibrinogen polymerization under the catalysis of thrombin, calcium ions, and/or factor XIII. Fibrinogen powder can dissolve in water to form a solution with low viscosity. Like gelatin, fibrin has excellent biocompatibility and biodegradability for various biomedical applications, such as hemostasia, wound healing, pharmacy, microencapsulated cell delivery, tissue engineering, and 3D bioprinting [104].

As stated above, layer-by-layer bioprinting is a logical choice for the manufacture of stratified organs, such as the liver, heart, kidney, and lung, containing multiple cell types. 3D bioprinting of stratified organ replacements depends on bioinks with appropriate rheological and cytocompatible properties. An obvious shortcoming of the fibrinogen solution for 3D organ bioprinting is that its viscosity is too low to be piled up in layers even in very high concentration (e.g., 4% w/v) [55]. After polymerization, the 3D printed constructs undergo further contraction or shrinkage for homogeneous or heterogeneous tissue formation. Structural stability is the major concern for pure fibrinogen as a bioink for 3D organ bioprinting.

The first pertinent fibrin 3D bioprinting was reported in 2007, in which fibrinogen was used as an additive of gelatin-based bioinks [105,106]. By blending fibrinogen with thermoresponsive gelatin solutions, high-resolution 3D constructs with excellent cell viability (i.e., ≈ 100%) have been obtained. During the 3D printing process, gelatin molecules provide fast gelation and immediate postprinting structural fidelity, while fibrinogen molecules ensure long-term mechanical stability upon thrombin polymerization. Physical blending and chemical crosslinking of fibrinogen/gelatin solutions can prevent breakdown and shrinkage of 3D printed constructs for a relatively longer period under physiological conditions (e.g., 37 °C).

More than 10 years later, Hinton and colleagues printed a Ca^2+^-mediated alginate/fibrinogen hydrogel into a gelatin slurry bath containing thrombin using a similar extrusion-based 3D printer and working principle [16,17]. When the alginate/fibinogen hydrogel was extruded from the 3D printer nozzle, it met the thrombin molecules in the bath and rapidly polymerized to solidify the Ca^2+^-mediated alginate/fibrinogen hydrogel. This is another double crosslinking or gelling path for 3D bioprinting with fibrinogen-containing composite bioinks and is termed as “bioplotting” (Figure 3). With different gelling paths, cells in the fibrinogen-containing hydrogel can be assembled in layers according to the predesigned CAD models.

Similarly, Burmeister and colleagues delivered ASCs via PEG–fibrin, i.e., PEGylated fibrin (FPEG) hydrogel, as an adjunct to meshed split-thickness skin grafts in a porcine model [107]. They showed that ASCs delivered in FPEG could enlarge the blood vessel size in a dose-dependent way, wherein FPEG acts as both a porous scaffold to prevent contraction and an ASC-delivery vehicle to accelerate angiogenesis.

For 3D organ bioprinting, the printing properties of fibrinogen solutions can be easily tailored by varying the proportion of its component in the composite bioinks [108,109,110,111]. Thermosensitive polymers, such as gelatin and agar, support extrusion-based 3D bioprinting of a wide range of natural polymers, including fibrinogen, chitosan, alginate, and hyaluronate. Succeeding (or tandem) chemical crosslinking and polymerization of fibrinogen-containing composite bioinks facilitates the printing results of the stratified cell-laden 3D constructs with increased stiffness and stabilization. For example, double gelation of gelatin/alginate/fibrinogen bioinks using both CaCl_2_ (for alginate ion crosslinking) and thrombin (for fibrinogen polymerization) is the most popular approach today for large-scale organ manufacturing (Figure 8) [30,36].

### 3.4. dECM

dECM is a mixture of natural polymers, such as collagen and glycosaminoglycans (GAGs), derived from decellularization of different animal tissues, such as the heart, skin, liver, and small intestinal submucosa. The decellularization process can be chemical, physical, enzymatic, or their combinations, and it plays an important role in the final dECM compositions and geometries. After decellularization, the ECM compositions and geometries of the original tissues can be highly preserved, providing specific micro/meso/macrobiophysical and biochemical environments for different cell lines. In particular, dECM is rich in cell growth factors and stem cell niches, which help to support special tissue generation. dECM has remarkable advantages in 3D bioprinting of customized tissues due to the preservation of personal ECMs.

Compared with gelatin and alginate, the viscosity of dECM solution is rather low. Like cell-laden bioinks, dECM-based solutions gel rapidly beyond 15 °C and form physically crosslinked hydrogels upon increase in the temperature and pH [112]. Like most other natural polymers, such as alginate, gelatin, and fibrinogen, dECM-based bioinks can form semipermeable substrates with entrapped viable cells and biomolecules for 3D organ bioprinting. Water, gases, nutrients, metabolites, and growth factors can infiltrate into the semipermeable substrates. Human ASC (hASC) viability can reach 90% on the 14th day in the 3D bioprinted dECM substrates [113].

During the 3D bioprinting process, dECM displays poor 3D printability, low accuracy (or resolution), and large shrinkage due to the low viscosity. Various attempts have been made to surmount the shortcomings. The attempts include chemical crosslinking and concomitantly printing dECM with other natural and/or synthetic polymers. For example, Pati and coworkers printed an adipose-derived dECM–PCL hybrid construct with ASCs encapsulated in the dECM using a two-head tissue building system. Cell viability in this hybrid construct was >90%, equal to that without PCL [114].

Though dECM has remarkable advantages for tissue-specific function preservation, it faces many challenges for complex 3D organ bioprinting due to the following reasons. Firstly, it is difficult to efficiently remove the immunogen existing in the allogeneic or xenogeneic dECMs to avoid immune responses of the host tissues. Secondly, more or less residual deoxyribonucleic acid (DNA) or nuclear materials retained in dECMs can affect the cellular behaviors of the adopted cells. Thirdly, the extremely weak mechanical property, poor shape fidelity, and rapid degradation rate are major issues for large vascularized and innervated organ construction [115,116,117].

## 4. Synthetic Polymers for 3D Organ Bioprinting

Synthetic polymers are human-made polymers produced by chemical reactions of monomers, which may be derived from petroleum oil. Their hydrogels are generally produced via bulk, solution, and inverse dispersion procedures. While the first two procedures are homogeneous, the inverse dispersion procedure is conducted in dispersed and continuous phases [118,119,120,121,122]. Among homogeneous polymerizations, the solution reaction is preferred due to better control of the heat of polymerization and hence the polymer properties. Most high-swelling synthetic polymeric hydrogels are produced in this way.

As a common characteristic, the glass transition temperatures of synthetic polymer solutions, such as poly(lactic-co-glycolic acid) (PLGA) and polyurethane (PU), are much lower than those of the abovementioned nature polymers. Temperatures of −20 to −40 °C are common for most biodegradable synthetic polymeric solutions to solidify. Meanwhile, very high temperatures, such as 100–200 °C, are necessary for biodegradable synthetic polymers to melt. For example, polylactic acid (PLA) has a melting temperature of 180 °C. Thus, synthetic polymers are comparatively bioinert and do not readily embody bioactive ingredients, such as cells and growth factors, directly for 3D bioprinting. This is mainly due to the fact that 3D printing processes of synthetic polymers often involve the use of organic solvents, heat, or poisonous crosslinkers, which may reduce the bioactivity of the ingredients [79,80,81].

Compared with natural polymers, synthetic polymers are prominent in their mechanical properties as their molecular weights can be regulated from low to ultrahigh according to the actual requirement. High molecular weights promote intermolecular interactions between the polymer chains and have better mechanical properties for in vitro pulsatile culture using peristaltic pump and in vivo implantation of the 3D printed constructs. Nevertheless, most synthetic polymer solutions, hydrogels, and scaffolds, such as PLGA, poly(glycolic acid) (PGA), poly(hydroxypropyl methacrylamide) (PHPMA), PU, PCL, PLA, and poly(methyl methacrylate) (PMMA), have poor cytocompatibilities due to the intrinsic bioinert characteristics, organic solvent usages, and stiff morphological/topological structures [123].

The lack of functional groups and structural complexity within synthetic polymers has limited their usage in 3D organ bioprinting. Till now, most synthetic polymers have been applied as supporting or sacrificing structures without directly contacting living cells. In this section, three synthetic polymers, namely, PEG, PLGA, and PU, with excellent 3D printability, in vivo tissue compatibility (or bioinertia), and structural stability for 3D organ bioprinting are introduced.

### 4.1. PEG

PEG, also named as polyoxyethylene or poly(ethylene oxide) (PEO), is a biocompatible, nonimmunogenic synthetic polyether that has been approved by the Food and Drug Administration (FDA) of the United States as a good candidate for cell encapsulation and other biomedical applications. It is a hydrophilic polymer with linear and branched structures. PEG can be crosslinked using physical, ionic, or covalent crosslinks. Two hydroxyl groups of PEG diol can be tailored into other functional groups (i.e., acrylate, thiol, and carboxyl) by physical, ionic, or covalent crosslinking, which makes PEG possess tunable mechanical properties for 3D bioprinting [124,125].

Unlike other synthetic polymers, PEG appears in solid state at room temperature with a molecular weight (Mw) about 1000 Da. PEG is water-soluble, and the viscosity depends on the Mw and on the amount of water. PEG itself cannot form hydrogel. The low viscosity of PEG solutions makes it impossible for use in extrusion-based 3D bioprinting. Acrylation or blending with other polymers is often necessary for the development of a PEG-containing hydrogel. For example, Gao and coworkers have extensively studied the properties of PEG–GelMA for inkjet-based bioprinting [126,127]. The resulting constructs could be used for hard tissue regeneration with a reasonable mechanical strength (e.g., compressive modulus: 1–2 MPa). GelMA in the bioink could promote mesenchymal stem cell differentiation into cartilage and bone tissues. This is due to the existence of natural gelatin component in the GelMA molecules. Similarly, the poly(ethylene glycol) diacrylate (PEG-DA) hydrogels photopolymerized by UV light could increase the mechanical properties and printing resolutions of 3D constructs but without enhanced biological functions [128].

Though PEG and its derivatives have been employed in 3D bioprinting, the lack of cell-adhesive domains and poor mechanical properties have seriously limited their application in 3D organ bioprinting. A great effort has been made to improve the biological and physiological functions of 3D printed constructs. One of the most widely used strategies is to incorporate bioactive molecules, especially arginyl–glycyl–aspartate (or Arg–Gly–Asp, RGD) peptide segments, in the bioinks to promote cell activities. For example, Villanueva and coworkers investigated the role of cell–matrix interactions by dynamically loading cells in a RGD-incorporated PEG 3D printing process. The attachment (e.g., adhesion) and differentiation capabilities of the cells, such as osteoblasts, mesenchymal stem cells, endothelial cells, and smooth muscle cells, on the 3D scaffold augmented in a dose-dependent manner. Through dynamic 3D bioprinting, the RGD-incorporated PEG bioink could obviously enhance the chondrocyte phenotype and ECM synthesis, indicating that cell–matrix interactions directly mediated cell activities [128]. Actually, RGD is a special peptide segment containing three amide linkages, which exists in all natural proteins, including collagen and its derivative gelatin. It has the same adhesive properties for cells as those of natural polymers.

### 4.2. PLGA

PLGA is a synthetic copolymer (i.e., linear aliphatic polyester) of lactic acid (α-hydroxy propanoic acid) and glycolic acid (hydroxy acetic acid), which has been approved by FDA for therapeutic devices. It is synthesized by means of ring-opening copolymerization of two different monomers: the cyclic dimers (1,4-dioxane-2,5-diones) of lactic acid and glycolic acid [129]. PLGA 75:25 has been identified as a commonly used copolymer with a composition of 75% lactic acid and 25% glycolic acid. The monomer lactic acid contains an asymmetric carbon atom and therefore has two optical isomers: l (+) lactic acid and d (61) lactic acid. It is widely distributed in all living creatures (such as animals, human bodies, plants, and microorganisms) as either an intermediate or an end product in carbohydrate metabolism. Meanwhile, glycolic acid occurs in nature to a limited extent.

PLGA can be hydrolyzed by breaking the ester linkages in its chains under the presence of water. The final degradation products of PLGA are either acidic monomers, such as lactic acid and glycolic acid, or innocuous salts, such as lactate (salt form of lactic acid) and glycolate (salt form of glycolic acid). It has been shown that the time required for the degradation of PLGA is related to the monomers’ ratio, which can be reflected in the molecular composition. The higher the content of glycolide units, the lower the time required for the degradation compared to the lactide predominant polymers [130].

PLGA dissolves in a wide range of organic solvents depending on its composition. Higher lactide-containing polymers dissolve in chlorinated solvents, whereas higher glycolide-containing polymers require the use of fluorinated solvents, such as 1,1,1,3,3,3-hexafluoroisopropanol. PLGA solutions show a typical glass transition temperature in the range of −40 to −60 °C.

Within the author’s own group, we have developed various low-temperature RP technologies to deposit synthetic PLGA solutions alone or with other polymers. Different material systems can be 3D printed together using double or multinozzle 3D bioprinters, resulting in hybrid constructs, such as PLGA–gelatin, PLGA–collagen, PLGA/hydroxyapatite–PLGA/hydroxyapatite/phosphralated chitosan, with strong mechanical properties, tunable biodegradabilities, and acceptable in vivo biocompatibilities (Figure 9) [20,50,131,132,133]. The 3D printed constructs have been extensively used for bone, cartilage, nerve, liver, and other large organ repair/regeneration/replacement/restoration. At the same time, the concept of vascularization and neuralization of large-scale 3D printed tissues has been adapted rapidly all over the world [134,135,136,137].

### 4.3. PU

PU is a family of synthetic polymers that are composed of organic units and joined by carbamate (i.e., urethane) links. PUs can be classified into two groups: biodegradable or nonbiodegradable. The traditional PUs are thermoresponsive (or thermoplastic, i.e., melt when heated) nonbiodegradable polymers that do not biodegrade when implanted in vivo. Historically, nonbiodegradable (or unbiodegradable) PU has been widely used in some biomedical fields due to its excellent mechanical and bioinert properties. Two examples of these biomedical applications are intravenous perfusion tubes and inanimate artificial hearts with strong mechanical strength [138,139]. However, these polymers cannot be printed using the existing 3D bioprinters.

In 2006, a brand new biodegradable elastomeric PU was developed by my group. This new PU is made of PEG and PCL monomers with excellent biocompatibility, biodegradability, bioprintability, and biostability for complex bioartificial organ manufacturing [136,137,138]. It can be 3D printed alone or with some other natural or synthetic polymers, such as gelatin, collagen, gelatin/alginate, etc. In one of our former studies, a hybrid hierarchical PU–cell/hydrogel construct was automatically created using an extrusion-based double-nozzle, low-temperature 3D printer (Figure 10) [28,49]. The PU mainly acted as a supportive template for cell, especially ASC, accommodation, growth, migration, proliferation, and differentiation. It is extremely useful for some antistress or antisuture applications, such as pulsatile culture of 3D printed hierarchical vascular and neural networks and in vivo anastomosis with the host vasculatures/neural networks [108,109,110]. The vascularized and innervated networks can be applied to 3D bioprinting of a variety of complex organs, such as the brain, heart, lung, and kidney.

Using similar techniques, a 3D printed PU/PEO scaffold was reported later in 2014. The PEO component could be used to increase the viscosity of the PU solution. The optimal ratio of PU/PEO for an extrusion-based FDM technology was 76/24. A resulting PU/PEO scaffold, containing an average pore size of 1–4 μm, was helpful to maintain cell morphology attached to it. Chondrocytes preferred to adhere onto the scaffold due to its hydrophilic property and suitable pore size [139]. In another study, neural stem cells (NSCs) attached well on the PU-based scaffold. The PU-based scaffold could provide NSCs with a proper environment to adhere, proliferate, and migrate on it and repair the disordered or damaged central nerve directly [140].

The control of multinozzle 3D printer parameters can provide a strong connection between different polymer systems, such as the PLGA solution–gelatin hydrogel, and PU solution–cell-laden gelatin-based hydrogel. Supportive synthetic polymers, such as PLGA and PU, and the ECM-like gelatin-based hydrogels can be adjusted to degrade at different time points during the organ construction and maturation stages. With proper selection of natural and synthetic polymer components, the 3D printed bioartificial organs can avoid all the risks of vascular rupture, stress shrinkage, immune rejection, and other negative reactions during the in vivo implantation stages [108,109,110]. These are all long-awaited breakthroughs in bioartificial organ manufacturing, as well as other pertinent hot research areas, such as tissue engineering, biomaterials, drug screening, organ transplantation, and pathological analysis, with respect to in vitro complex organ automatic building processes and in vivo defective/failed organ repair/regeneration/replacement/restoration applications.

## 5. Challenges and Perspectives

Despite remarkable progress achieved in 3D bioprinting over the last decade, there are still a number of challenges that remain in the manufacturing of off-the-shelf bioartificial organs for clinical applications. These challenges include (1) the extreme difficulties in establishment of an axial anastomosed vasculature, which anatomically mimics those in a natural organ, providing the incorporated cells with water, gas, and nutrients and removing metabolites from the cells; (2) the limited sophisticated multinozzle 3D printers, which are capable of assembling as many homogeneous and heterogeneous cells along with different polymers in a way similar to those in a natural organ; (3) the unmatched physical, biological, and physiological properties of 3D printed vascular, neural, and lymphatic networks for defective/failed organ restoration; (4) the uncertain roles of superfluous stem cells, growth factors, and other bioactive agents existing in the bioartificial organs for in vivo implantation [141,142,143].

In the future, the demand for 3D organ bioprinting, both for production in large amounts and small quantities as well as individual manufacture (i.e., customized), will definitely grow. More research needs to focus on mimicking the complex anatomical, material, and biological aspects of each human organ to be replaced, especially the hierarchical vascular, neural, and lymphatic networks for fluid and bioactive signal transportation. It is imperative to develop new sophisticated polymers with updating 3D bioprinters to recapitulate more complex micro-, meso-, and macro-3D milieus of natural organs with respect to geometrical features, material constituents, and environmental factors [144,145,146].

In the future, suitable choice of natural and synthetic polymers for 3D bioprinting of each bioartificial organ will still play a critical role. Much more integration of the existing natural and synthetic polymers for additional physiological functionality realization in a special natural organ is still on the way. High cell viability is a crucial prerequisite for complex 3D organ bioprinting with a large volume of homogeneous and heterogeneous cells. It is necessary to further clarify the special usages of various polymers for different tissue incorporation. Harsh 3D bioprinting conditions should be avoided or overcome through additional approaches. Patient-specific cells, especially stem cells and ECMs (or dECMs), are preferable to eliminate the immune reactions [147,148,149,150,151,152].

In the future, much more work needs to be done on stem cell extraction, culture, and induction. Because a human organ is composed of at least two types of cells, specific architectural design and multiple cell assembling are essential for each of the layer-by-layer 3D building processes. Sequential induction of stem cell differentiation in a predefined 3D construct needs to be normalized in a suitable laboratory [108,109,110]. In particular, ASCs have set an outstanding example for their vascularization and neuralization capabilities in 3D printed constructs. Further optimization of growth factors for different stem cell inductions needs to be carried out and standardized.

## 6. Concluding Remarks

3D organ bioprinting is a new, high-level interdisciplinary field that requires the integration of talents of many fields of science and technology, such as cell biology, computer, materials, information, chemistry, mechanics, engineering, manufacturing, and medicine. Bioartificial organs with tailored biological, biophysical, biochemical, and physiological properties can be 3D bioprinted through predesigned geometrical structures, biomaterial components, and processing parameters. Both natural and synthetic polymers have played essential and ubiquitous roles in successful 3D organ bioprinting technologies. Some natural polymer hydrogels, such as gelatin, fibrinogen, and dECM, have excellent cytocompatibilities due to their inorganic solvents, functional groups, and mild gelation conditions. Cells and bioactive agents can be incorporated in polymer solutions or hydrogels directly without significantly changing their viabilities and bioactivities. Biodegradable synthetic polymers, such as PLGA and PU, hold antisuture and antistress capabilities, which can be used as supporting structures for vascular and neural network strengthening in 3D printed constructs. The integration of natural and synthetic polymers using multinozzle 3D bioprinters in this author’s own laboratory have paved all the rugged ways for bioartificial organ manufacturing with several series of successful 3D printing models. Further studies are still needed to develop more successful bioinks for 3D bioprinting of each human organ with a whole spectrum of physiological functions, such as multiple cell/ECM types, special geometrical shapes, and antisuture/antistress capabilities. Advanced polymer-based multidisciplinary efforts will reap much greater benefits in 3D organ bioprinting and will virtually replace failed/defective human organs in the near future.

## Figures and Tables

**Figure 1 micromachines-10-00814-f001:**
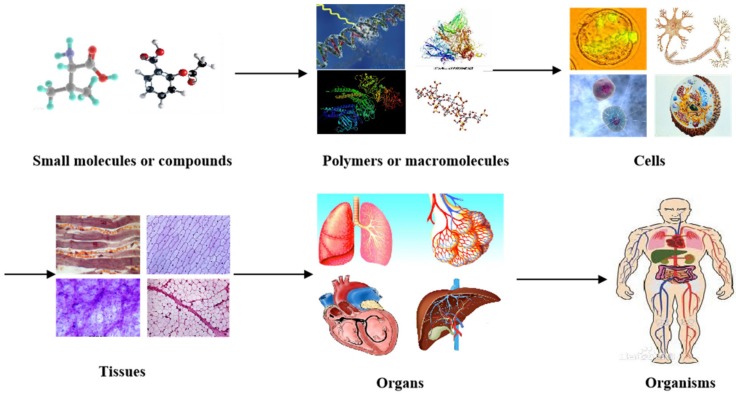
Different levels of materials (or molecules) existing in the human body, from small organic molecules or compounds to larger polymers (i.e., macromolecules or biomacromolecules), cells, tissues, organs, and systems in organisms.

**Figure 2 micromachines-10-00814-f002:**
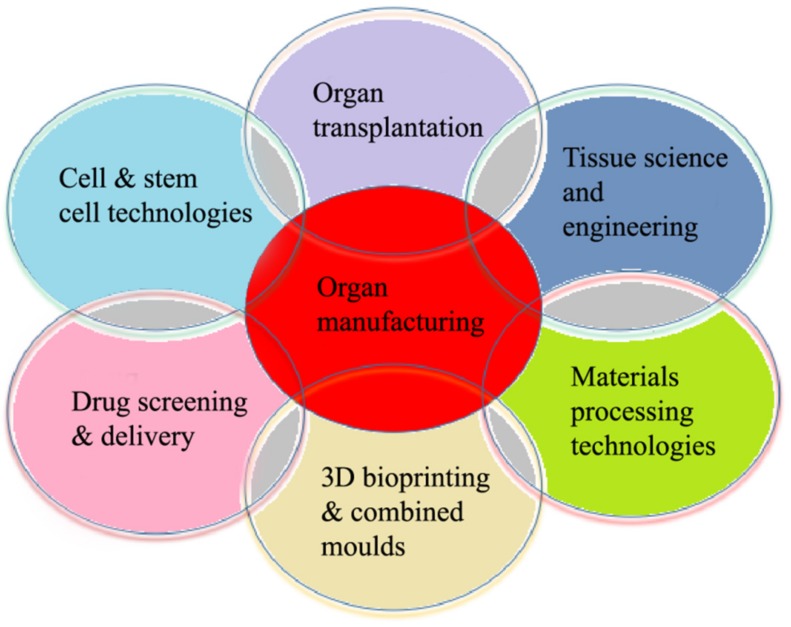
Relationships of organ manufacturing, including three-dimensional (3D) bioprinting, with other pertinent fields of science and technology.

**Figure 3 micromachines-10-00814-f003:**
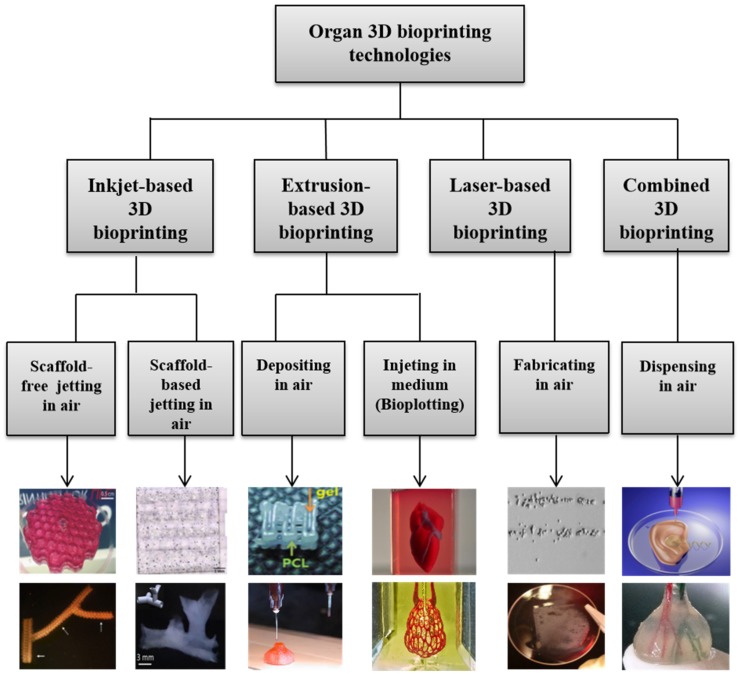
Graphical description of 3D bioprinting types [10,11,12,13,14,15,16,17,18,19,20]. Images reproduced with permission from [10,11,12,13,14,15,16,17,18,19,20].

**Figure 4 micromachines-10-00814-f004:**
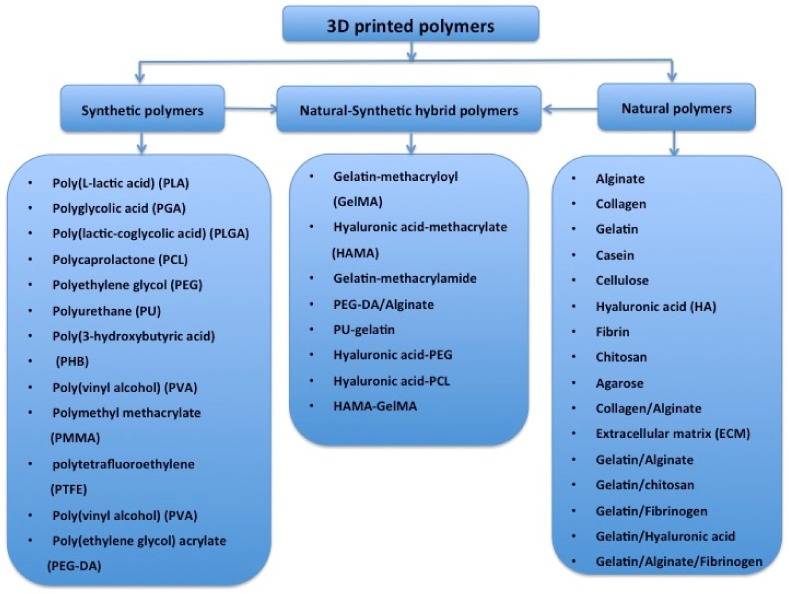
Polymers that have been used for tissue and 3D organ printing.

**Figure 5 micromachines-10-00814-f005:**
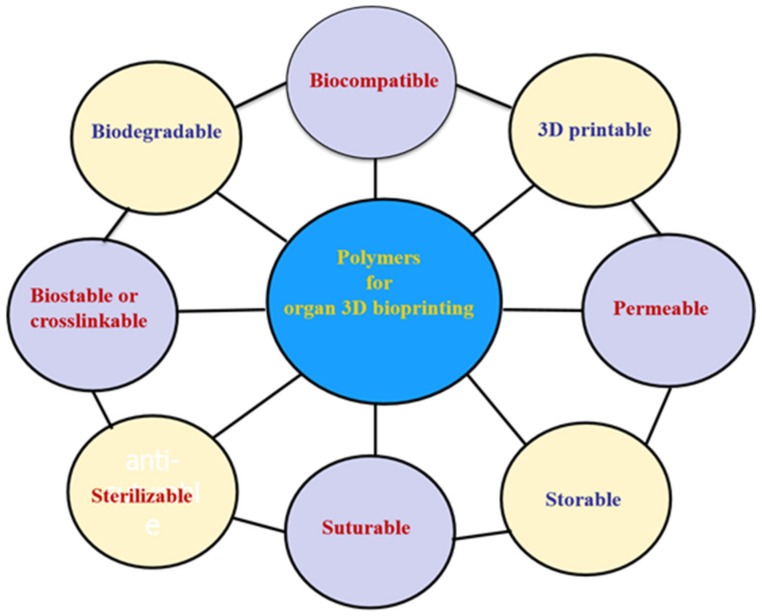
Basic requirements for selecting a polymer for 3D bioprinting of bioartificial organs.

**Figure 6 micromachines-10-00814-f006:**
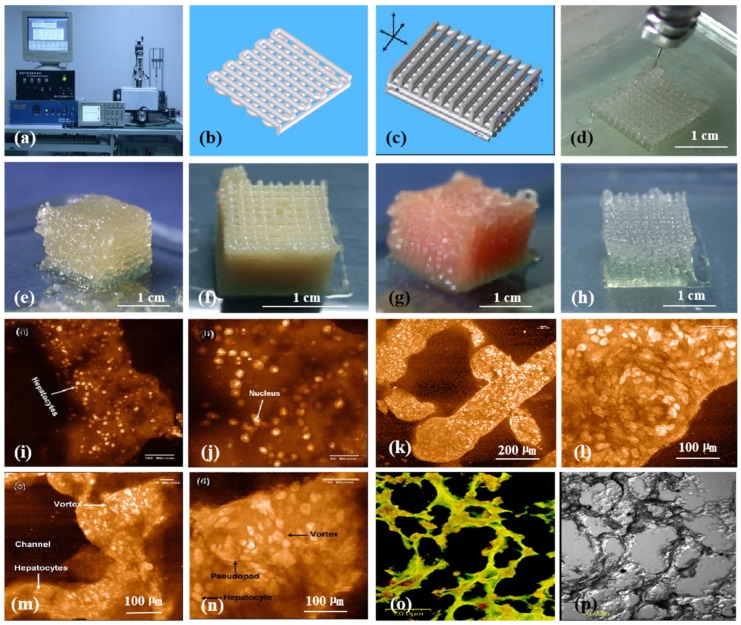
3D bioprinting of chondrocytes, cardiomyocytes, hepatocytes, and adipose-derived stem cells (ASCs) into living tissues/organs using a pioneering 3D bioprinter made in Prof. Wang’s laboratory at Tsinghua University: (**a**) the pioneering 3D bioprinter; (**b**) schematic description of a cell-laden gelatin-based hydrogel being printed into an one-layer grid lattice using the 3D bioprinter; (**c**) schematic description of the cell-laden gelatin-based hydrogel being printed into large-scale 3D construct using the 3D bioprinter; (**d**) 3D printing process of a chondrocyte-laden gelatin-based construct; (**e**) a grid 3D construct made from a cardiomyocyte-laden gelatin-based hydrogel; (**f**) hepatocytes encapsulated in a gelatin-based hydrogel after 3D printing; (**g**) hepatocytes in a gelatin-based hydrogel after 3D printing; (**h**) a gelatin-based hydrogel after 3D printing; (**i**) hepatocytes in a 3D printed construct after a certain period of in vitro culture; (**j**) a magnified photo of (i); (**k**) the shape of the hepatocyte-laden grid construct maintained well after certain period of in vitro culture; (**l**) a magnified photo of (k), showing hepatocytes formed vortex in the hydrogel; (**m**) hepatocytes in a 3D printed construct after a longer period of in vitro culture; (**n**) a magnified photo of (m), showing the vortex structure was still there; (**o**) immunostaining of the hepatocyte-laden 3D construct after certain period of in vitro culture, showing new hepatic tissue formed in the gelatin-based hydrogel with close cell-cell connection or tight junction; (**p**) a dark-field microscopy of (o). Images reproduced with permission from [7,22].

**Figure 7 micromachines-10-00814-f007:**
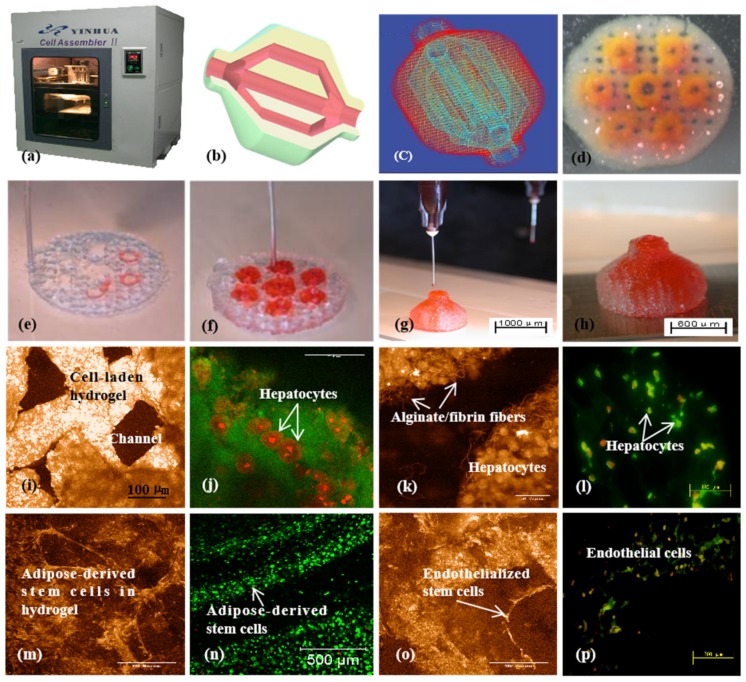
A large-scale 3D printed bioartificial organ with vascularized liver tissue constructed through the double-nozzle 3D bioprinter created in Prof. Wang’s laboratory at Tsinghua University: (**a**) the double-nozzle 3D bioprinter; (**b**) a computer-aided design (CAD) model containing a branched vascular network; (**c**) a CAD model containing the branched vascular network; (**d**) a few layers of the 3D bioprinted construct containing both ASCs encapsulated in a gelatin/alginate/fibrin hydrogel and hepatocytes encapsulated in a gelatin/alginate/chitosan hydrogel; (**e**–**h**) 3D printing process of a semielliptical construct containing both ASCs and hepatocytes encapsulated in different hydrogels; (**i**–**l**) hepatocytes encapsulated in the gelatin-based hydrogels after 3D bioprinting and different periods of in vitro cultures; (**m**–**p**) ASCs encapsulated in the gelatin-based hydrogels after 3D bioprinting and different periods of in vitro cultures as well as growth factor inductions. Images reproduced with permission from [15,32].

**Figure 8 micromachines-10-00814-f008:**
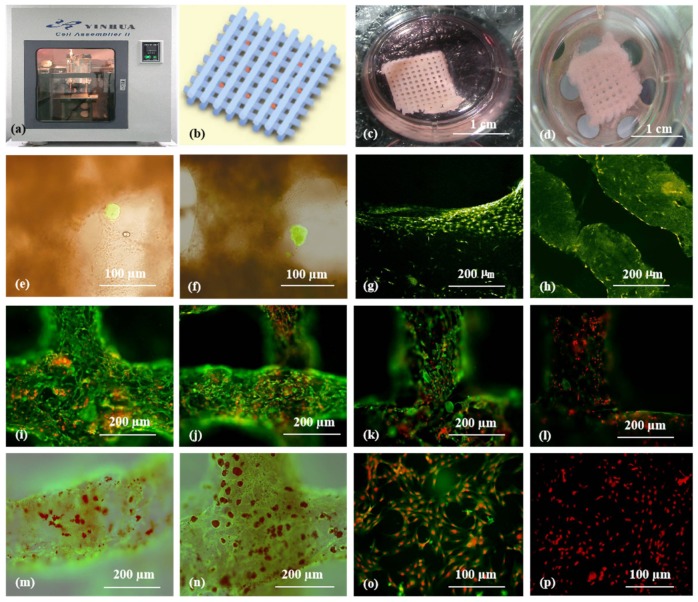
3D bioprinting of ASC-laden gelatin/alginate/fibrin hydrogel for organ manufacturing in Prof. Wang’s laboratory at Tsinghua University: (**a**) a pioneering double-nozzle 3D bioprinter made in this laboratory; (**b**) schematic description of the cell-laden gelatin/alginate/fibrin hydrogel and pancreatic islets being printed into a grid construct using the 3D bioprinter; (**c**) a large-scale 3D printed grid construct containing ASC-laden gelatin/alginate/fibrin hydrogel cultured in a plate; (**d**) a grid ASC-laden gelatin/alginate/fibrin construct after being cultured for one month; (**e**) a multicellular construct after three weeks of culture, containing both ASCs encapsulated in the gelatin/alginate/fibrin hydrogel before epidermal growth factor (EGF) engagement and relatively integrated pancreatic islets seeding in the predefined channels (immunostaining with anti-insulin in green); (**f**) some envelopes of the islets were broken after one month of culture; (**g**) immunostaining of the 3D construct with mAbs for CD31^+^ cells (i.e., mature endothelial cells from the ASC differentiation after three days of culture with EGF added in the culture medium) in green, having a fully confluent layer of endothelial cells (i.e., endothelium) on the surface of the predefined channels; (**h**) a vertical image of the 3D construct showing the fully confluent endothelium (formed from endothelial cells) and the predefined go-through channels; (**i**) immunostaining of the 3D construct with mAbs for CD34^+^ cells (i.e., endothelial cells) in green and pyrindine (PI) for cell nuclei (nucleus) in red; (**j**) immunostaining of the 3D construct with mAbs for CD34^+^ endothelial cells in green and PI for cell nuclei (nucleus) in red after three days of culture without EGF added in the culture medium; (**k**) immunostaining of the 3D construct with mAbs for CD31^+^ endothelial cells in green and PI for cell nuclei (nucleus) in red after three days of culture with EGF added in the culture medium; (**l**) a control of (k), immunostaining of the 3D construct with mAbs for CD31^+^ endothelial cells differentiated from the ASCs in green and PI for cell nuclei (nucleus) in red after three days of culture without EGF added in the culture medium; (**m**) immunostaining of the 3D construct with mAbs for CD31^+^ cells in green and Oil Red O staining for adipocytes in red, showing both the heterogeneous tissues coming from the ASC differentiation after a cocktail growth factor engagement (i.e., on the surface of the channels, the endothelium coming from the ASCs differentiation after being treated with EGF for 3 days; deep inside the gelatin/alginate/fibrin hydrogel, the adipose tissue coming from the ASC differentiation after being subsequently treated with insulin, dexamethasone, and isobutylmethylxanthine (IBMX) for another three days); (**n**) a control of (m), showing all the ASCs in the 3D construct differentiated into target adipose tissue after three days of treatment with insulin, dexamethasone, and IBMX but no EGF; (**o**) immunostaining of two-dimensional (2D) cultured ASCs with mAbs for CD31^+^ endothelial cells in green and PI for cell nuclei (nucleus) in red after three days of culture with EGF added in the culture medium; (**p**) immunostaining of 2D cultured ASCs with mAbs for CD31^+^ endothelial cells in green and PI for cell nuclei (nucleus) in red after three days of culture without EGF added in the culture medium. Images reproduced with permission from [36].

**Figure 9 micromachines-10-00814-f009:**
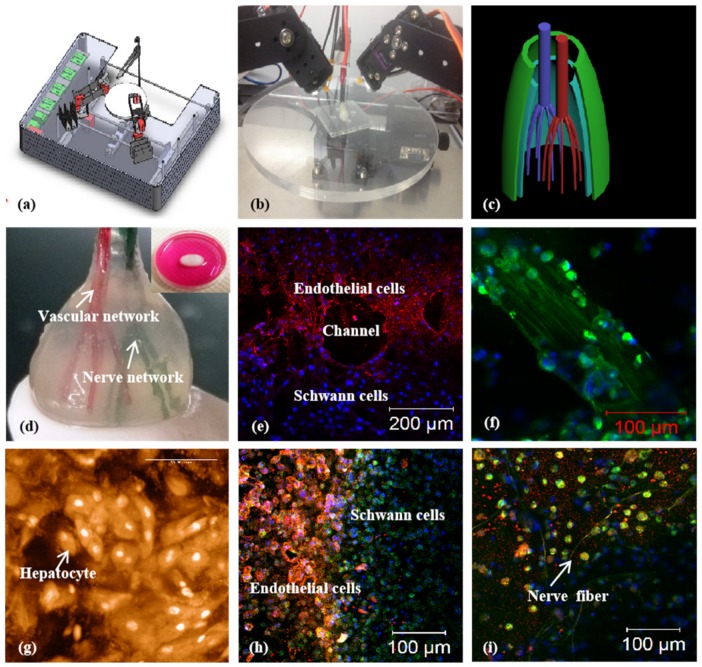
A combined four-nozzle 3D organ bioprinting technology created in Prof. Wang’s laboratory at Tsinghua University in 2013 [20,50]: (**a**) equipment of the combined four-nozzle 3D organ bioprinter; (**b**) working state of the combined four-nozzle 3D organ printer; (**c**) a CAD model representing a large-scale vascularized and innervated hepatic tissue; (**d**) a semielliptical 3D construct containing a poly(lactic-co-glycolic acid) (PLGA) overcoat, a hepatic tissue made from hepatocytes in a gelatin/chitosan hydrogel, a branched vascular network with fully confluent endothelialized ASCs on the inner surface of the gelatin/alginate/fibrin hydrogel, and a hierarchical neural (or innervated) network made from Shwann cells in the gelatin/hyaluronate hydrogel; the maximal diameter of the semiellipse can be adjusted from 1 mm to 2 cm according to the CAD model; (**e**) a cross section of (d), showing the endothelialized ASCs and Schwann cells around a branched channel; (**f**) a large bundle of nerve fibers formed in (d); (**g**) hepatocytes underneath the PLGA overcoat; (**h**) an interface between the endothelialized ASCs and Schwann cells in (d); (**i**) some thin nerve fibers.

**Figure 10 micromachines-10-00814-f010:**
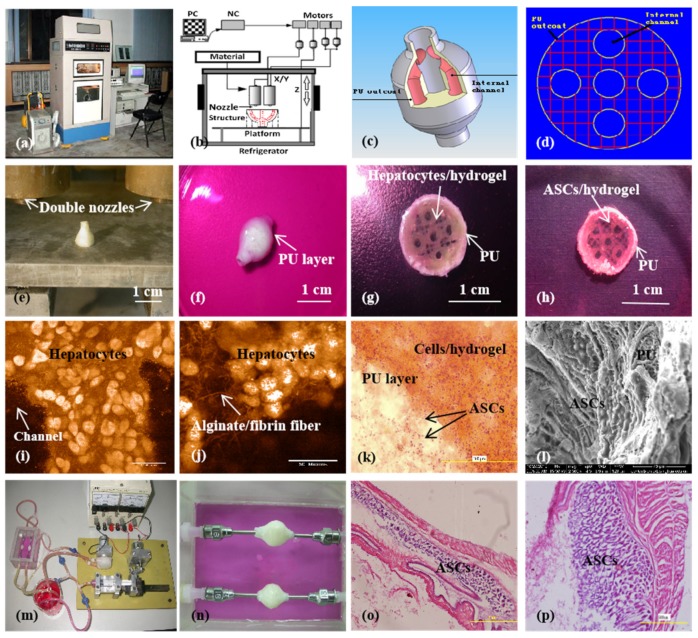
A large-scale 3D printed complex organ with vascularized liver tissue constructed through the double-nozzle low-temperature 3D bioprinter created in Prof. Wang’s laboratory at Tsinghua University: (**a**) the double-nozzle low-temperature 3D bioprinter; (**b**) working principle of an elliptical hybrid hierarchical polyurethane-cell/hydrogel construct built via the double-nozzle low-temperature 3D bioprinter; (**c**) a CAD model containing the branched vascular network; (**d**) a cross section of the CAD model containing five sub-branched channels; (**e**) working platform of the 3D bioprinter containing two nozzles and the 3D printed semielliptical hybrid hierarchical polyurethane-cell/hydrogel construct; (**f**) an elliptical sample containing both a cell-laden natural hydrogel and a synthetic polyurethane (PU) overcoat; (**g**) several layers of the elliptical sample in the middle section containing a hepatocyte-laden gelatin-based hydrogel and a PU overcoat; (**h**) several layers of the elliptical sample in the middle section containing an ASC-laden gelatin-based hydrogel and a PU overcoat; (**i**) hepatocytes encapsulated in the gelatin-based hydrogel; (**j**) a magnified photo of (i), showing the alginate/fibrin fibers around the hepatocytes; (**k**) ASCs encapsulated in the gelatin-based hydrogel growing into the micropores of the PU layer; (**l**) ASCs on the inner surface of the branched channels; (**m**) pulsatile culture of two elliptical samples; (**n**) two samples cultured in the bioreactor; (**o**) static culture of the ASCs encapsulated in the gelatin-based hydrogel; (**p**) pulsatile culture of the ASCs encapsulated in the gelatin-based hydrogel. Images reproduced with permission from [28,49].

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
