# Peer review of "Advanced Polymers for Three-Dimensional (3D) Organ Bioprinting"

_micromachines, 2019, doi:10.3390/mi10120814_

Round 1

Reviewer 1 Report

1)Figure 2 needs to be modified with other pertinent science and technologies for example tissue engineering and regenerative medicine are interchangeable term and it uses biomaterial. The figure can be modified in accordance with line 57 and make it clear the role of the intersection of all branches of sciences in organ printing.

2) The author claims that “we have solved all the bottleneck problems” is a very bold author who needs to reconsider this statement and citations are missing for lines 70-81.

3) The author needs to be sure in lines 173-174 about G/M ratio or M/G ratio.

4) The author should include the PCL and GelMA as they are the most commonly used polymer for 3D bioprinting with good printability and tissue compatibility.

5) Some citations are missing.

Author Response

Figure 2 needs to be modified with other pertinent science and technologies for example tissue engineering and regenerative medicine are interchangeable term and it uses biomaterial. The figure can be modified in accordance with line 57 and make it clear the role of the intersection of all branches of sciences in organ printing.

Response: Figure 2 has been modified accordingly, and the error “biomterial” has been corrected.

The author claims that “we have solved all the bottleneck problems” is a very bold author who needs to reconsider this statement and citations are missing for lines 70-81.

Response: Yes, this is a very bold author who has no better way to prevent scientific cheats, misdeeds, crimes, etc., as well as resource wastes and sacrifices.

The author needs to be sure in lines 173-174 about G/M ratio or M/G ratio.

Response: That sentence has been changed.  

The author should include the PCL and GelMA as they are the most commonly used polymer for 3D bioprinting with good printability and tissue compatibility.

Response: GelMA has been included in the text (page 11, line 283-286). PCL has been mentioned in page 14, line 394-397. Additionally, PCL has been presented in tails in the author’s another article.

5) Some citations are missing.

Response: More references have been added [18-40].

Reviewer 2 Report

In this review the author reports on polymers used in Bioprinting. The paper is interesting since it reports a wide overview on the different polymers used. I have no particular issue on the work except few remarks, here listed

1 There are few typos in the paper, please check. One is in figure 2 (biomterials)

2 in the part of synthetic polymers, the author reports the sol-gel transition temperature of synthetic polymers but usually it is called glass transition temperature. Then, melting point of the polymers is related to biodegradation of the polymer, but actually it is not like this (for instance PLA is biodegradable but it has a melting temperature of 180 °C)

3 the author mentions the low viscosity of PEO/PEG. Actually PEG is solid at RT with a molecular weight about 1000 Da and the viscosity depends on the Mw and on the amount of water.

4 In PU section is reported that "traditional PU are thermoresponsive (or thermosettings)(i.e. melt when heated)". Thermosets do not melt when heated, the right word is thermoplastics.

I have a single big concer, regarding the similaritis of this review with previous reviews ( https://doi-org.ezproxy.biblio.polito.it/10.1177%2F0963689718809918 , https://doi.org/10.3390/polym10111278)

Author Response

In this review the author reports on polymers used in Bioprinting. The paper is interesting since it reports a wide overview on the different polymers used. I have no particular issue on the work except few remarks, here listed.

Response: Thanks a lot for the faithful comments!

1 There are few typos in the paper, please check. One is in figure 2 (biomterials).

Response: The typos have been corrected.

2 in the part of synthetic polymers, the author reports the sol-gel transition temperature of synthetic polymers but usually it is called glass transition temperature. Then, melting point of the polymers is related to biodegradation of the polymer, but actually it is not like this (for instance PLA is biodegradable but it has a melting temperature of 180 °C).

Response: The statements have been revised (page 14, line 413-418].

3 the author mentions the low viscosity of PEO/PEG. Actually PEG is solid at RT with a molecular weight about 1000 Da and the viscosity depends on the Mw and on the amount of water.

Response: The statements have been revised (page 14, line 446-448].

4 In PU section is reported that "traditional PU are thermoresponsive (or thermosettings)(i.e. melt when heated)". Thermosets do not melt when heated, the right word is thermoplastics.

Response: The error has been corrected (page 16, line 518).

I have a single big concer, regarding the similaritis of this review with previous reviews ( https://doi-org.ezproxy.biblio.polito.it/10.1177%2F0963689718809918 , https://doi.org/10.3390/polym10111278)

Response: The author cannot find the former article. There are no repeat or similar statements of this article with the second one.

Round 2

Reviewer 1 Report

Being bold will not prevent scientific cheats or misdeeds, but presenting work in scientific way will prevent this for sure.Thank you

Author Response

Comment: Being bold will not prevent scientific cheats or misdeeds, but presenting work in scientific way will prevent this for sure.Thank you

Response: The comment is great! Thanks a lot!
